# *Rhynchophorus palmarum* (Linnaeus, 1758) (Coleoptera: Curculionidae): Guarani-Kaiowá indigenous knowledge and pharmacological activities

Kellen Natalice Vilharva[1], Daniel Ferreira Leite[1], Helder Freitas dos Santos[1], Katia Ávila Antunes[1], Paola dos Santos da Rocha[1], Jaqueline Ferreira Campos[1], Claudiane Vilharroel Almeida[2], Maria Lígia Rodrigues Macedo [2], Denise Brentan Silva[3], Caio Fernando Ramalho de Oliveira[1,2], Edson Lucas dos Santos[1], Kely de Picoli Souza[1]*

1 Research Group on Biotechnology and Bioprospecting Applied to Metabolism, Federal University of Grande Dourados, Dourados, Brazil, 2 Protein Purification Laboratory and its Biological Functions, Federal University of Mato Grosso do Sul, Campo Grande, Brazil, 3 Laboratory of Natural Products and Mass Spectrometry, Federal University of Mato Grosso do Sul, Campo Grande, Brazil

* kelypicoli@gmail.com

**Data Availability Statement:** All relevant data are within the manuscript and its Supporting information files.

## Abstract

Zootherapy is a traditional secular practice among the Guarani-Kaiowá indigenous ethnic group living in Mato Grosso do Sul, Brazil. My people use the oil extracted from larvae of the snout beetle *Rhynchophorus palmarum* (Linnaeus, 1758) to treat and heal skin wounds and respiratory diseases. Based on this ethnopharmacological knowledge, the chemical composition and antioxidant, antimicrobial, and healing properties of *R. palmarum* larvae oil (RPLO) were investigated, as well as possible toxic effects, through *in vitro* and *in vivo* assays. The chemical composition of the RPLO was determined using gas chromatography coupled with mass spectrometry. The antioxidant activity of RPLO was investigated through the direct 2,2-diphenyl-1-picrylhydrazyl (DPPH) radical scavenging assay, and the antimicrobial activity was evaluated against Gram-positive and Gram-negative bacteria that are pathogenic to humans. The healing properties of RPLO were investigated by performing a cell migration assay using human lung fibroblasts (MRC-5), and the toxicity was analyzed, *in vivo*, using a *Caenorhabditis elegans* model and MRC-5 cells, *in vitro*. RPLO contains 52.2% saturated fatty acids and 47.4% unsaturated fatty acids, with palmitic acid (42.7%) and oleic acid (40%) representing its major components, respectively. RPLO possesses direct antioxidant activity, with a half-maximal inhibitory concentration ($IC_{50}$) of 46.15 mg.ml$^{-1}$. The antimicrobial activity of RPLO was not observed at a concentration of 1% (v/v). RPLO did not alter the viability of MRC-5 cells and did not exert toxic effects on *C. elegans*. Furthermore, MRC-5 cells incubated with 0.5% RPLO showed a higher rate of cell migration than that of the control group, supporting its healing properties. Taken together, RPLO possesses direct antioxidant activity and the potential to aid in the healing process and is not toxic toward *in vitro* and *in vivo* models, corroborating the safe use of the oil in traditional Guarani-Kaiowá medicine.

**Funding:** The authors received no specific funding for this work.

**Competing interests:** The authors have declared that no competing interests exist.

## Introduction

Traditional knowledge (TK) is the result of the secular experience of these communities living among the local biodiversity, their relationship with the environment, and their use of different species for their benefit [1–4]. The indigenous Guarani-Kaiowá community is located in the Central-West region of Brazil. Although the practice of zootherapy is common among the Guarani-Kaiowá, records of the species used, the purposes of treatments, and the forms of drug preparation are scarce. Some of the secular knowledge related to zootherapy is transmitted to the younger generation by the *Maxuypy*, the matriarch of the family. In this manner, I, Kellen Natalice Villarva, a biologist, and one of the authors of this study, received part of the knowledge of my ethnicity from my grandmother.

Insects are routinely used in indigenous communities for medicinal purposes [5]. The use of *Mbuku*, the name given to the beetle *Rhynchophorus palmarum* (Linnaeus, 1758) (Coleoptera: Curculionidae), in its larval stage is part of the Guarani-Kaiowá TK, whether in its sacred tales, in its rituals or for food and medicinal purposes. The oil extracted from the integument of *R. palmarum* larvae is used by other peoples of South America in food and as a treatment for respiratory diseases and skin infections [6–9]. While the use of animal fat in the treatment of diseases by indigenous people is part of TK, the literature describes the antioxidant properties of unsaturated and saturated fatty acids that compose the triglycerides of animals and attributes these compounds to the ability to neutralize oxidative damage [10], both after topical use and oral ingestion [11, 12].

In the Guarani-Kaiowá indigenous community, *R. palmarum* larvae are used as a source of oil for the treatment of wounds to accelerate healing and prevent infection. This oil is also used to treat respiratory ailments through its application to the chest and back. This study presents knowledge and teachings about the insects used by the Guarani-Kaiowá. The record of indigenous TK and the tale of the creation of *Mbuku* are described in S1 File. Given the scarcity of information regarding the chemical composition and pharmacological properties of *R. palmarum* oil, we recorded the traditional process for collecting larvae and obtaining *R. palmarum* larvae oil (RPLO) according to the Guarani-Kaiowá TK. In addition to determining the chemical features of RPLO, we investigated its antioxidant, antimicrobial, and healing properties, as well as its possible toxicity.

## Materials and methods

### Ethics statement

Part of this study was carried out in Takuara village, located in the municipality of Juti, Mato Grosso do Sul, Brazil. The residents who participated of process of larvae collection has given written informed consent to publish case details. Authorization for the collection of *R. palmarum* larvae was granted by Mrs. Júlia Cavalheiro, the matriarch and leader of the Takuara village. The field studies did not involve endangered or protected species.

### Obtaining *R. palmarum* larvae oil

In the village, the process for obtaining oil from *R. palmarum* larvae is performed during the day, mainly by the mothers and grandmothers of the family. As part of the ritual of paying respect to the protectors and the owners of the forest and the animals, the collectors pray before entering the forest. The larvae are collected by opening the stipe of previously felled palm trees with the aid of tools such as a knife and ax (Fig 1). After larvae collection, the oil extraction is performed exclusively by women, who are gathered in a group of a maximum of three people. Importantly, as part of the Guarani-Kaiowá ritual, the entire process must be

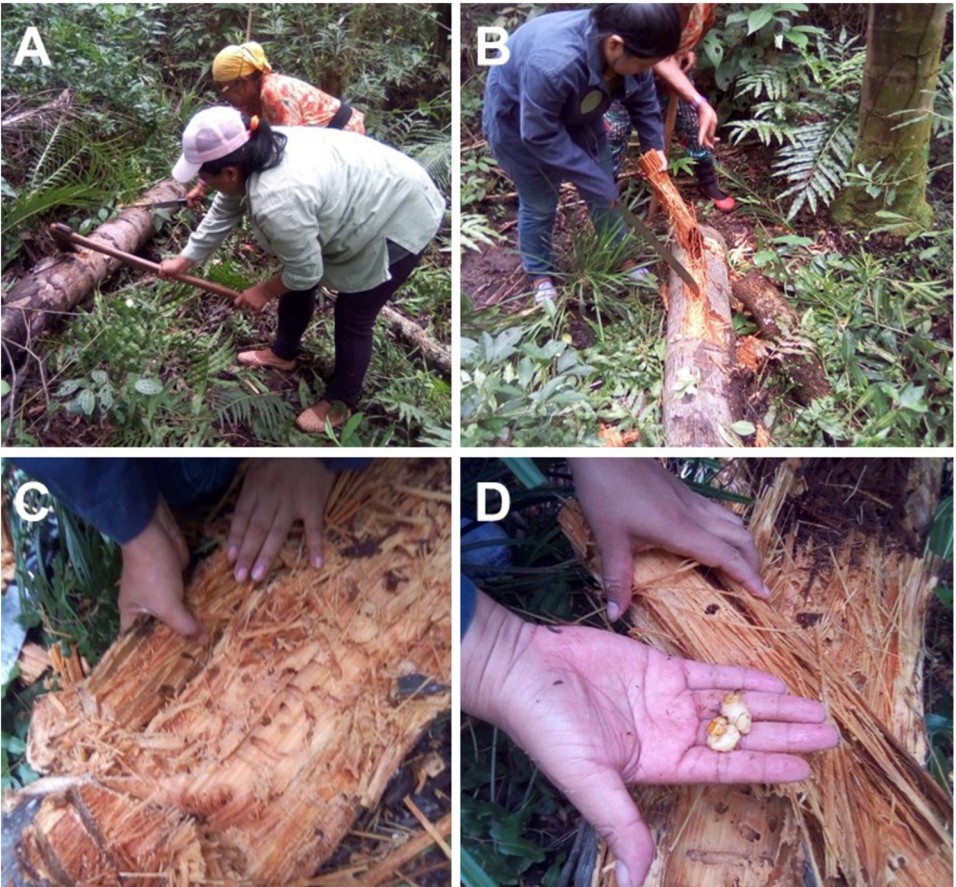

**Fig 1. Collection of *R. palmarum* larvae.** (A) Palm trees felled; (B) palm branches being opened; (C) fibers where the larvae develop, and (D) collection of the larvae.

performed in silence. The larvae are placed in a container that is placed on the fire; at which time the oil is extracted from the larval carcass. In the village, the extracted oil may be stored together with or without the larval carcasses in closed containers to be used when needed.

For this study, *R. palmarum* larvae were collected in Takuara village (22˚43'10.1"S 54˚38'10.9"W), which is located in the municipality of Juti, Mato Grosso do Sul, Brazil, in February 2019. During the collection, the larvae were placed in containers containing palm stipe as a substrate and transported to the Laboratory of Biotechnology and Bioprospecting Research applied to Metabolism (GEBBAM), Federal University of Grande Dourados (UFGD). A total of 30 larvae were collected, washed with distilled water, dried with paper towels, weighed, and frozen (-20˚C) until the time of oil extraction.

The procedure for extracting RPLO reproduced the traditional method performed in Takuara village. The larvae were thawed overnight in a refrigerator (4˚C) and cut into pieces to allow the extravasation of the internal contents. Then, the larvae were heated on a glass plate at a controlled temperature (150˚C) for 15 min. During the heating, the larvae were stirred with a glass rod to prevent them from adhering to the glass. At the end of the process, the oil was separated from the carcasses by filtration, its volume was determined, and 15 ml of oil divided into aliquots were obtained (Fig 2). For use in the assays, the oil was aliquoted into 2-ml microtubes and stored at room temperature in the dark.

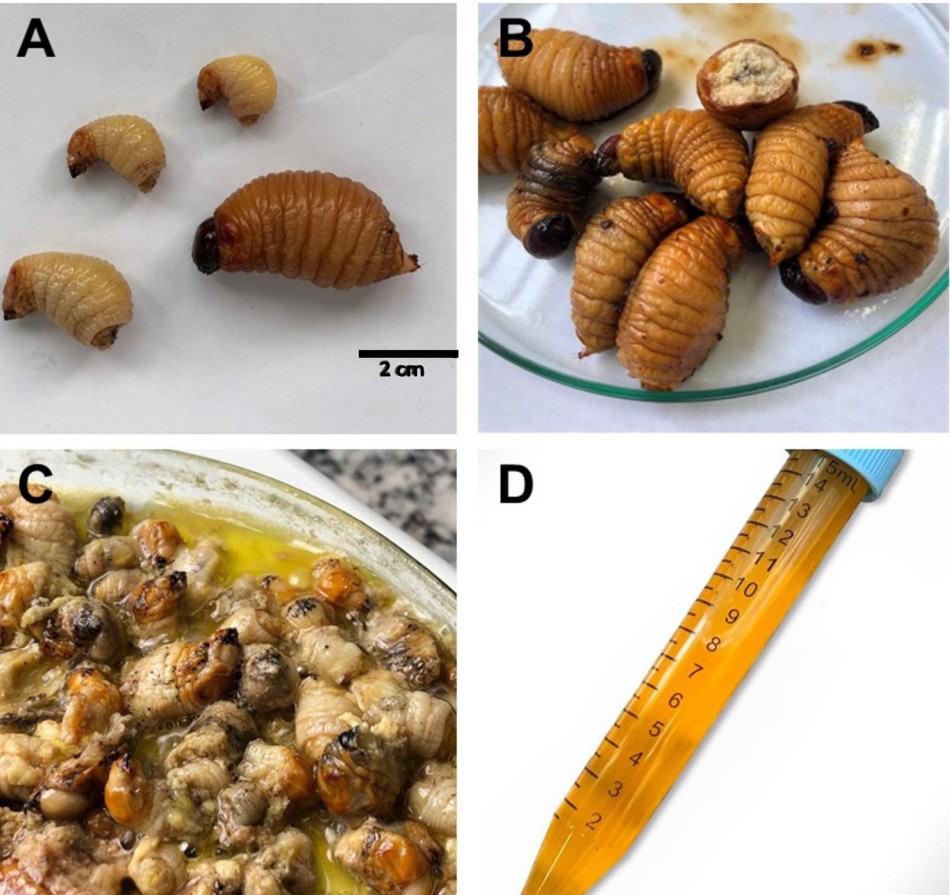

**Fig 2. Process of obtaining oil from *R. palmarum* larvae.** (A) Larvae of different sizes; (B) thawing of the larvae; (C) heating process, where it is possible to observe the release of oil next to the larvae integument and (D) oil collected at the end of the heating step.

## Determination of the chemical composition of RPLO

To evaluate the fatty acid composition of RPLO the transesterification reaction was performed using the method described by Koohikamali et al. [13]. First, 2.5 ml of a 1 M sodium methoxide (Sigma-Aldrich, São Paulo, Brazil) solution was added to 10 mg of the oil and incubated at 70˚C for 30 min. The solution was incubated at room temperature for 3 h. Next, 1.5 ml of ultrapure water were added to the mixture, and the free fatty acids were extracted with 1 ml of hexane, followed by stirring and incubation of the mixture for 15 min. The hexane fraction was collected after the formation of the water: hexane phases. Hexane extraction was performed three times, for a total volume of 3 ml. Subsequently, the hexane fractions were concentrated to a final volume of 1 ml, and 1 μl was analyzed using gas chromatography coupled with mass spectrometry (GC-MS).

The oil analyzes were performed in a gas chromatograph Shimadzu QP2010 coupled to a mass spectrometer, operating in electronic ionization (EI), using 70 eV ionization energy. An Rtx-5MS chromatographic column (30 m x 0.25 mm, 0.25 mm thick) was used with helium as a carrier gas at a pressure of 91.2 kPa and a flow rate of 1.4 ml.min$^{-1}$. The split ratios were 70 and 10 for the transesterification reaction analyzis. For the analysis of the transesterification reaction, the temperature programming was 60˚C for 3 min, 60–310˚C with increments of

$6°C. min^{-1}$ and $310°C$ for 13 min. The databases WILEY 7 and NIST 11 were applied to identify the compounds present in the oil.

## Antioxidant activity of RPLO

The antioxidant capacity of RPLO was evaluated using the direct 2,2-diphenyl-1-picrylhydrazyl (DPPH) radical scavenging method described by Gupta and Gupta in 2011, with minor modifications [14]. In this assay, different concentrations of RPLO ($13–105$ mg.ml$^{-1}$), diluted in chloroform, were added to 3 ml of DPPH solution (0.11 mM DPPH prepared in chloroform). Then, the solution was homogenized and incubated for 30 min at room temperature, protected from light. Subsequently, the absorbance of the samples was measured at 517 nm in a T70 UV/VIS spectrophotometer (PG Instruments Limited, Leicestershire, United Kingdom). The Trolox compound was used as a reference antioxidant. The capture percentage was calculated using the equation:

$$DPPH\ uptake\ percentage = 1 - \frac{Abs\ sample}{Abs\ negative\ control}\ x\ 100$$

Abs sample refers to the mixture of DPPH with different concentrations of RPLO and Abs negative control is the absorbance of DPPH solution (0.11 mM DPPH in chloroform). The concentration of RPLO required to inhibit 50% (IC$_{50}$) the formation of DPPH radical was also determined. The antioxidant activity was determined through the average of three independent experiments carried out in triplicate ($n = 9$).

## Antimicrobial activity of RPLO

The antimicrobial activity of RPLO was evaluated by the broth microdilution technique [15], performed in 96-wells microplates. The bacterial inoculum was prepared using the direct growth method. Colonies isolated on Mueller Hinton agar (MHA) were placed in 0.9% sterile NaCl solution until turbidity equal to 0.5 on the Mac Farland scale ($1x10^8$ CFU.ml$^{-1}$), determined in a turbidimeter (MS Tecnopon, São Paulo, Brazil). The bacterial suspension was diluted 1:20 (v/v), thus reaching a concentration of $5x10^6$ CFU.ml$^{-1}$. Gram-positive strains *Staphylococcus aureus* ATCC 35983, *Staphylococcus epidermidis* ATCC 35984, *Staphylococcus haemolyticus* ATCC 49453, *Staphylococcus saprophyticus* ATCC 29970, and Gram-negative *Acinetobacter baumannii* ATCC 19606, *Enterobacter aerogenes* ATCC 1960, *Enterobacter cloacae* ATCC 13047, *Escherichia coli* ATCC 35218, *Klebsiella oxytoca* ATCC 13182, *Proteus mirabilis* ATCC 12453, *Salmonella enterica* ATCC 51741, and *Serratia marcescens* ATCC 13880 were used.

The diluted bacterial suspension ($5x10^6$ CFU.ml$^{-1}$) was added to the wells of the microplate containing the Mueller Hinton (MH) culture medium, containing 1% oil (v/v). The bacterial concentration in each well was $5x10^5$ CFU.ml$^{-1}$. An aliquot of the RPLO was prepared in a 0.9% sterile NaCl solution and used to prepare working solutions containing 10% of the RPLO. In each well, 80 μl of MH broth, 10 μl of the oil working solution, and, subsequently, 10 μl of the bacterial inoculum were added, making a volume of oil in the well of 1%. As a positive control of growth inhibition, 80 μl of MH broth, 10 μl of the antibiotic chloramphenicol at a concentration of 4 μg.ml$^{-1}$ and 10 μl of the bacterial inoculum were added to the well. The concentration of chloramphenicol used in the study represents the minimum bactericidal concentration (BMC). As a negative control of growth inhibition, 80 μl of MH broth, 10 μl of sterile 0.9% NaCl solution, and 10 μL of the bacterial inoculum were added to the well. The experiments were carried out in triplicate ($n = 9$) through three independent assays.

The microplate was incubated at 37°C under constant agitation and monitored at 30 min intervals in a Multiskan GO microplate reader (Thermo Fisher Scientific, São Paulo, Brazil) at 595 nm. After 18 h of the experiment, the percentage of bacterial growth inhibition was calculated using the last reading of the exponential growth phase, according to the equation:

$$\text{Percentage of growth inhibition} = 1 - \frac{\text{Abs sample}}{\text{Abs negative control}} \; x \; 100$$

Abs sample refers to samples containing RPLO in different concentrations and Abs negative control is the absorbance of the negative growth inhibition control. The minimum inhibitory concentration (MIC) was that which inhibited 100% of bacterial growth.

## Cell culture, viability, and migration

The human lung fibroblast cell line MRC-5 was cultured in DMEM (Sigma-Aldrich, São Paulo) supplemented with 10% fetal bovine serum and 1% antibiotics (5 mg.ml$^{-1}$ penicillin, 5 mg.ml$^{-1}$ streptomycin, and 10 mg.ml$^{-1}$ neomycin; Gibco/Invitrogen, Minneapolis, MN, USA). The cells were cultured in an incubator with a humidified atmosphere containing 5% $CO_2$ at 37°C.

Cell viability was evaluated using a colorimetric assay with the reagent 3-(4,5-dimethylthiazol-2-yl)-2,5-diphenyltetrazolium bromide (MTT). For this assay, $5x10^3$ cells were plated in each well of a 96-well microplate and incubated with different concentrations of RPLO (0.1 to 0.5% of the total volume) for 24 h. Next, 100 μl of MTT (0.5 mg.ml$^{-1}$, prepared in DMEM medium) were added to each well, followed by another incubation for 4 h at 37°C. Subsequently, the medium containing MTT was removed, and 100 μl of dimethyl sulfoxide (DMSO) was added to the wells to solubilize the formazan crystals. The absorbance was measured at 630 nm using a SpectraMax 250 microplate reader (Molecular Devices, San Jose, California, USA). The inhibition of cell viability was calculated using the equation:

$$\text{Cell viability}(\%) = \frac{\text{Abs treated cells}}{\text{Abs positive control cells}} \; x \; 100$$

Abs treated cells refer to the samples incubated with RPLO and Abs positive control is the absorbance of the cells incubated only with culture medium. MRC-5 fibroblasts were cultured in 24-well microplates at a density of $8x10^4$ cells/well with DMEM supplemented with 10% fetal bovine serum for 24 h to investigate the ability of RPLO to stimulate cell migration. After the cells had adhered to the plates, the culture medium was aspirated, and a vertical streak was generated in each well with a 1,000-μl pipette tip. Then, the wells were washed with PBS and photographed. Five hundred microliters of DMEM containing 0.5% RPLO were added to each well, and the plate was incubated at 37°C for 24 h. For the control treatment, only the DMEM medium was added, without the addition of RPLO. At the end of the 24-h period, the culture medium was aspirated, and the wells were photographed. Three images of each well were acquired: one in the center of the well and the others at equidistant positions between the middle and the edges of the well, as shown in S1 File. The areas to be photographed were marked with a permanent pen to ensure that the images captured at 0 and 24 h were acquired from the same position. The images captured at 0 (after the creation of the wound with the pipette tip) and 24 h were analyzed with ImageJ software and used to calculate the area without cells and consequently the cell migration rate. This assay was performed in triplicate (n = 9), and the results are presented as the wound areas in mm$^2$, according to the method described by Chen et al. [16].

### Viability test in a *Caenorhabditis elegans* model

The nematode *C. elegans* wild-type N2 was used for the *in vivo* experiments. The nematodes were maintained at a temperature of 20˚C, reared in Petri dishes containing nematode growth medium (NGM), and fed with *Escherichia coli* OP50 bacteria [17].

The RPLO acute toxicity test was performed using the method described by Bonamigo et al. [18]. Ten young adult nematodes (L4) were transferred to each well of a 96-well microplate and exposed to 200 μl of M9 medium containing different RPLO concentrations ranging from 1 to 4.5%. The animals were incubated at 20˚C for 24 and 48 h. As a negative control, the nematodes were incubated with 200 μl of M9 culture medium. After the incubation period, the viability of the nematodes was evaluated by assessing their sensitivity to touch with a platinum loop. The nematodes were evaluated using a Motic SMZ-140 & W10X/23 stereomicroscope. The results were reported as the means of three independent tests performed in triplicate (n = 30).

### Statistical analysis

The data were expressed as mean ± standard error of the mean (SEM). Prior to the statistical analysis, Levene´s test of homogeneity of variances was used to assume that variances are equal across groups or samples and Kolmogorov-Smirnov test was used to assess the normality of the data. The mean values of the cell migration rate were analyzed by One-way analysis of variance (ANOVA) followed by Dunnett's post-test. In in vivo assays, a t-test was used to determine differences between groups. All analyses were performed using GraphPad Prism 5 software. The results were considered to be statistically significant at $p \leq 0.05$.

## Results

### Yield and chemical composition of RPLO

The process for obtaining the oil was reproduced according to the Guarani-Kaiowá indigenous method. The determination of the oil yield was based on the mean of three extractions. The mean oil content was 0.15 ml.g$^{-1}$ of the larva. At the end of the RPLO extraction process, the oil was yellow and had an odor and viscosity characteristic of the oil processed at Takuara village. Thus, the experiments were continued.

Saturated and unsaturated fatty acids were identified in the RPLO, which are shown in Table 1. The determination of the chemical composition revealed that RPLO comprises saturated (SFA) and unsaturated (UFA) fatty acids, with a slightly higher SFA content. Palmitic (16:0), myristic (14:0) and stearic (18:0) acids were the major SFAs. Oleic (18:1, ω-9), palmitoleic (16:1, ω-7), and linoleic (18:2, ω-6) acids were identified as UFAs, representing the omega-6, -7 and -9 families, respectively.

### Antioxidant activity of RPLO

The antioxidant activity of RPLO was determined using the DPPH radical scavenging assay. The antioxidant activity of RPLO was concentration-dependent, generating a dose-response curve. Table 2 shows the IC$_{50}$ values of RPLO and of the antioxidant Trolox, which was used as a reference antioxidant because it is an analog of vitamin E, a fat-soluble vitamin.

### Antimicrobial activity of the RPLO

The antibacterial activity of RPLO was evaluated against Gram-positive and Gram-negative bacteria to determine the minimum inhibitory concentration (MIC) and minimum bactericidal concentration (MBC). As shown in Table 3, the oil did not exhibit antimicrobial activity

**Table 1. Fatty acids identified in the RPLO.**

| Peak | RT (min) | Compound | Number of carbons: unsaturation position | % |
|---|---|---|---|---|
| 1 | 19.1 | Dodecanoic acid (lauric acid) | 12:0 | 0.1 |
| 2 | 22.9 | Tetradecanoic acid (myristic acid) | 14:0 | 5.4 |
| 3 | 26.0 | Hexadecenoic acid (palmitoleic acid) | 16:1 ($\omega$-7) | 6.3 |
| 4 | 26.4 | Hexadecanoic acid (palmitic acid) | 16:0 | 42.7 |
| 5 | 29.0 | 9,12-octadecadinoic acid (linoleic acid) | 18:2 ($\omega$-6) | 1.1 |
| 6 | 29.1 | 9-octadenoic acid (oleic acid) | 18:1 ($\omega$-9) | 40.0 |
| 7 | 29.5 | Octadecanoic acid (stearic acid) | 18:0 | 3.1 |
| 8 | 32.3 | Eicosanoic acid (arachidic acid) | 20:0 | 0.9 |
| **SFA** | | | | 52.2 |
| **UFA** | | | | 47.4 |
| **Total** | | | | **99.6** |

RT: Retention time; SFA: saturated fatty acids; UFA: unsaturated fatty acids.

against any bacteria up to the maximum concentration tested (1%). At higher concentrations, the oil interfered with the absorbance reading of the assay. For this reason, higher concentrations were not used.

## Effect of RPLO on the viability of MRC-5 cells

According to Fig 3, the average viability of the control cells and the cells incubated with RPLO, in concentrations between 0.1 and 0.5%, did not differ statistically. RPLO concentrations greater than 0.5% were not analyzed due to changes in turbidity of the medium, which influenced the photometric reading. For this reason, all experiments were carried out with RPLO at a maximum concentration of 0.5%.

## Effect of RPLO on the migration rate of MRC-5 cells

The effect of RPLO on the migration rate of MRC-5 cells was observed at 0 and 24 h after the incubation with RPLO. As shown in Fig 4A, at time 0 h, the control and RPLO-treated wells had similar areas, being the empty areas resulted from the removal of the cells. After 24 h, a higher reduction in the exposed area was observed in the wells treated with 0.5% RPLO compared to the control. The cell migration rate was statistically increased by 60% in the wells treated with RPLO compared to the wells receiving the control treatment (Fig 4B).

## Toxicity of RPLO in a *C. elegans* model

None of the RPLO concentrations was able to affect the *C. elegans* viability (Fig 5), reinforcing that the tested concentrations of RPLO did not exert acute toxic effects *in vivo*.

**Table 2. Determination of the $IC_{50}$ values for RPLO and Trolox, the reference antioxidant, in the direct DPPH radical scavenging assay.**

| Compound | $IC_{50}$ (mg.ml$^{-1}$) |
|---|---|
| RPLO | 46.31 $\pm$ 3.02 |
| Trolox | 3.88 $\pm$ 0.57 |

**Table 3. Determination of the Minimum Inhibitory Concentration (MIC) and Minimum Bactericidal Concentration (MBC) of RPLO.** The antibiotic chloramphenicol was used as a positive control for the inhibition of bacterial growth.

|  | *Bacteria* | *MIC* | *MBC* | *Chloramphenicol (μM)* |
|---|---|---|---|---|
| **Gram-positive** | *Staphylococcus aureus* ATCC 35983 | N.D. | N.D. | 24 |
|  | *Staphylococcus epidermidis* ATCC 35984 | N.D. | N.D. | 12 |
|  | *Staphylococcus haemolyticus* ATCC 49453 | N.D. | N.D. | 12 |
|  | *Staphylococcus saprophyticus* ATCC 29970 | N.D. | N.D. | 24 |
| **Gram-negative** | *Acinetobacter baumannii* ATCC 19606 | N.D. | N.D. | 24 |
|  | *Enterobacter aerogenes* ATCC 13048 | N.D. | N.D. | 12 |
|  | *Enterobacter cloacae* ATCC 13047 | N.D. | N.D. | 12 |
|  | *Escherichia coli* ATCC 35218 | N.D. | N.D. | 24 |
|  | *Klebsiella oxytoca* ATCC 13182 | N.D. | N.D. | 12 |
|  | *Proteus mirabilis* ATCC 12453 | N.D. | N.D. | 24 |
|  | *Salmonella enterica* ATCC 51741 | N.D. | N.D. | 12 |
|  | *Serratia marcescens* ATCC 13880 | N.D. | N.D. | 12 |

N.D. Not determined at the concentration tested.

## Discussion

The TK is a cultural element of the social, religious, economic, and healthy relationships of indigenous peoples. Within this knowledge, the use of animals for the treatment of diseases is

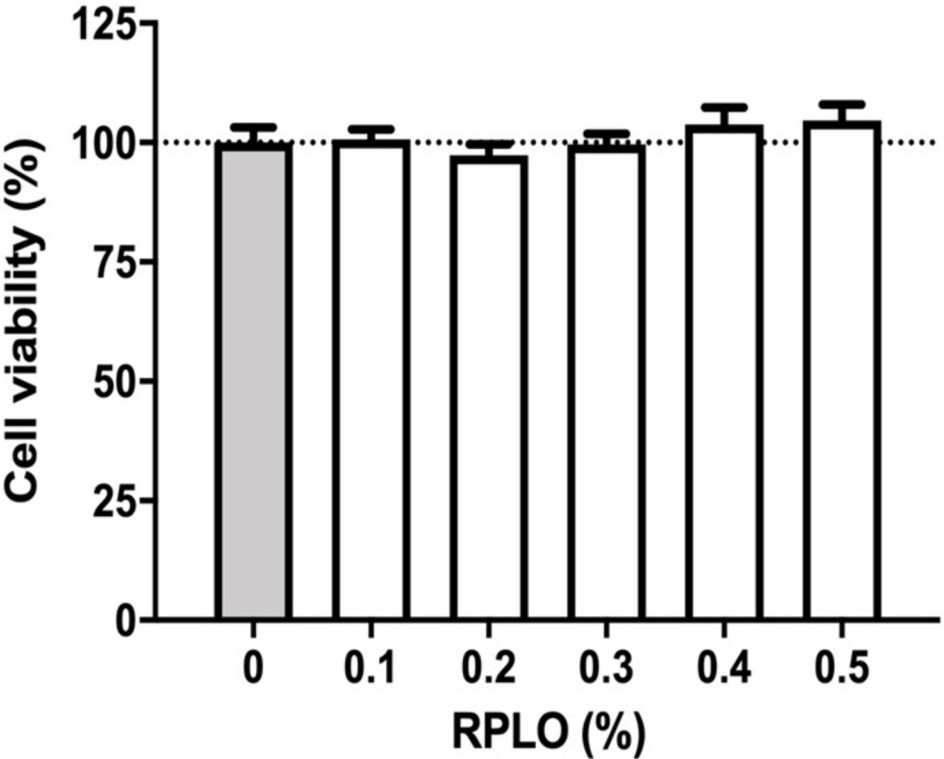

**Fig 3. Viability of MRC-5 cells incubated with different concentrations of RPLO for 24 h.** The graph shows the means ± SEM of three independent experiments performed in triplicate (n = 9). The observed changes were not statistically significant (t-test).

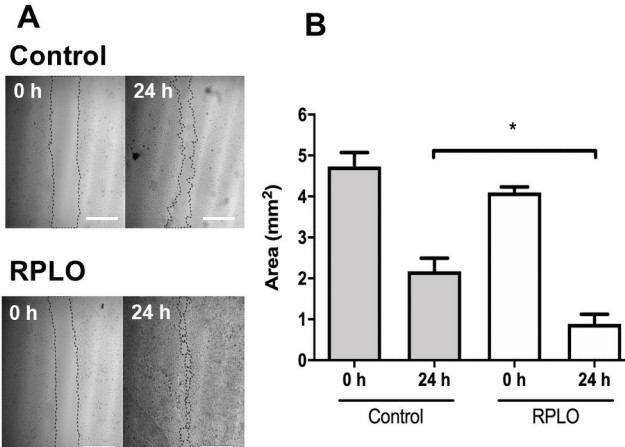

**Fig 4. The effect of RPLO on the migration rate of MRC-5 cells.** (A) Representative image from assay. The wound areas were determined at 0 h and after 24 h of incubation. Control cells were incubated with culture media. Cells incubated with 0.5% of oil were named RPLO. Scale bar is 1 mm. (B) The graph shows the means ± SEM of three independent experiments performed in triplicate (n = 9). An asterisk (*) indicate statistically significant differences between mean values (one-way ANOVA, Dunnett's post-test, p <0.05).

a practice that has been maintained for centuries from generation to generation, particularly through the oral transmission of this knowledge. Changes in the lifestyle of indigenous communities have hindered their abilities to maintain customs that favor orality while increasing access to other means of recording, transmitting, and revisiting this knowledge. The inclusion of indigenous people in universities has allowed knowledge holders to investigate the potential pharmacological properties of traditional medicines. Thus, a Guarani-Kaiowá woman (the first author of this manuscript) records for the first time the knowledge of the use of RPLO by her community. The present study described the antioxidant activities and healing potential of RPLO and the absence of toxic effects, consistent with its traditional use in wound healing.

Traditionally, indigenous communities maximize the rational consumption of natural resources. An example of this practice is the method used to obtain *R. palmarum* larvae. Part of the stipes of palm trees felled for the construction of houses is left on the forest floor, at which time the *R. palmarum* beetles lay their eggs. Later, the women return to the forest to collect the larvae. In addition to the use of stipes for building houses and obtaining larvae, the

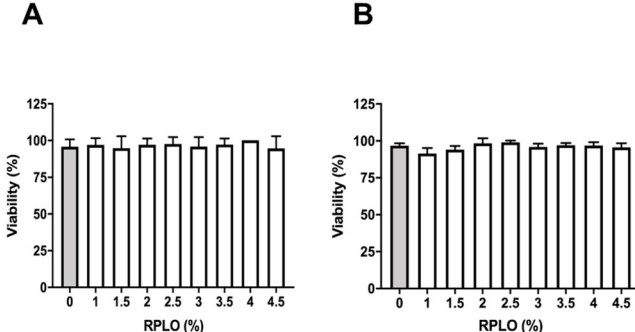

**Fig 5. Acute toxicity of RPLO in the C. elegans experimental model in vivo.** Viability was recorded at (A) 24 h and (B) 48 h of treatment. The graph shows the means ± SEM of three independent experiments performed in triplicate (n = 90). The observed changes were not statistically significant (t-test).

fibers of the palm leaves are used to make nets and as roofs for houses. The traditional method of obtaining *R. palmarum* larvae only occurs in villages with a considerable available area of natural forest. Thus, access to local biodiversity allows the establishment of a permanent ecological balance with adequate management and sustainable extraction to meet the demands of the community.

Most of the drugs on the market are derived from plants [19]. However, the list of extracts and molecules isolated from animals is increasing [20, 21]. RPLO is used by the Guarani-Kaiowá community and other indigenous communities for food [7] and medicinal purposes [6]. The chemical composition of RPLO revealed the presence of saturated [palmitic acid (C16:0), 42.7%], and unsaturated fatty acids [oleic acid (C18:1) [ω-9] 40%] in proportions similar to those observed by Due et al. [22]. Once the chemical composition of RPLO was known, we investigated the pharmacological properties attributed to its major constituents.

Palmitic acid (C16:0), the main SFA of RPLO, is common in the human diet [23]. However, the excess intake of SFAs increases serum low-density lipoprotein (LDL) cholesterol levels [24], which is associated with obesity [25], diabetes [26], heart disease, and cancer [27].

UFAs have attracted the interest of researchers in the areas of skin cosmetics and food due to their numerous known benefits to human health. The intake of UFAs increases the amount of high-density lipoprotein (HDL) cholesterol [28], promoting a reduction in LDL levels. Oleic acid (C18:1, ω-9, 40% of RPLO), which is abundant in olive oil used in the Mediterranean diet and RPLO, has been described to regulate immune and anti-inflammatory functions [29, 30], reduce depressive symptoms [31], and exert positive effects on the prevention and treatment of cardiovascular, autoimmune and metabolic diseases, and cancer [32]. Dietary supplementation of diabetic mice with palmitoleic acid (C16:1 [ω-7] 6.3% of RPLO) promoted a reduction in body weight, hyperglycemia, and hypertriglyceridemia [33], and is an important positive modulator of glucose uptake [34]. In the study by Souza et al. [35], the intake of palmitoleic acid promoted a reduction in hepatic steatosis and the levels of inflammatory mediators and improved peripheral insulin resistance in mice, attenuating the production of hepatic glucose induced by a high-fat diet [36]. Linoleic acid (C18:2 [ω-6] 1.1% of RPLO), although it is a minor constituent of RPLO, is a precursor of important endogenous fatty acids, such as arachidonic and eicosapentaenoic acids, which have shown cardioprotective properties [37, 38] and may attenuate the inflammatory process involved in metabolic syndrome, as well as cell adhesion, apoptosis, and cancer [39, 40].

UFAs, such as those found in RPLO, are components of the three layers of the skin, acting as an important physical barrier [41], or in tissue repair processes after mechanical damage [42]. Fatty acid deficiency caused skin inflammation similar to atopic dermatitis in hairless mice fed a special diet [43]. Oleic acid (ω-9) is widely used in cosmetic emulsions because it possesses emollient properties, restoring the oiliness of dry and flaky skin [44]. Besides, oleic acid enhances the permeation of molecules from the stratum corneum of the epidermis towards deeper layers of this tissue without exerting toxic or irritating effects [45]. Based on these characteristics, several UFAs have been investigated as potential percutaneous carriers of drugs, such as antidepressants, anti-inflammatory agents, and treatments for Alzheimer's disease [46, 47].

The generation of lipid mediators from linoleic acid, such as arachidonic acid and, subsequently, eicosanoids such as prostaglandins, thromboxanes, and leukotrienes, participates in the division and differentiation of epithelial tissue cells [48] and hemorrhage and hemostasis [44]. Therefore, the participation of essential fatty acids is critical for the maintenance of skin health and in repair processes favoring tissue regeneration [49]. The addition of RPLO to human fibroblast cultures increased the cell migration rate compared to the control treatment. An increased fibroblast migration rate was also observed after supplementation of the medium with a mixture of vegetable oils [50]. In addition to the positive effects on the migration rate of

fibroblasts, Guidoni et al. [51] observed that a mixture of vegetable oils promoted an inflammatory response controlled by the modulation of the levels of pro-inflammatory mediators and cytokines (TNF-α and IL-6). This controlled response promoted an improvement in wound tissue healing. These results corroborate the healing properties of other oils, such as pequi and buriti oil, which stimulate the healing of experimental skin wounds in rats [52, 53].

Another beneficial property of RPLO is its antioxidant activity. Extensive exposure to ultraviolet radiation contributes to premature skin aging [54], increasing the risk of carcinogenic processes in this tissue [55]. The observed antioxidant activity of RPLO helps to reduce oxidative damage in the skin, particularly the damage responsible for photoaging. Oleic acid has been added to photoprotective and tanning emulsions due to its photoprotective properties, promoting skin regeneration and reducing the damage caused by excessive exposure to sunlight [56, 57].

In addition to the photoprotective properties, the topical application of oils decreases the trauma during dressing replacement, prevents dehydration of the wounded tissue, and functions as a barrier against microorganism invasion into the temporally unprotected tissue [58]. Based on the results of *in vitro* assays, RPLO does not exert antimicrobial effects on the evaluated bacteria. The absence of direct antimicrobial actions was also described for the oil extracted from the snake *Spilotes pullatus* (Linnaeus, 1758) [59], in contrast to the oil extracted from the Brazilian boa *Boa constrictor* Linnaeus, 1758, [60]. Thus, further studies of the antimicrobial properties of oils are encouraged.

The antioxidant properties of the UFAs, namely, oleic [12], linoleic [61], and stearic [12] acids, are very important both for topical application and for ingestion because oxidative damage occurs at both the tissue and systemic levels [62]. Studies examining the antioxidant activity of oils obtained from insects is scarce in the literature [63–66]; however, the antioxidant activity of RPLO was similar to vegetable oils shown to be beneficial to health, such as sunflower, macadamia, linseed, and chia oils [67].

The consumption of foods considered antioxidants is recommended. However, excess intake of antioxidants may promote pro-oxidant effects [68, 69]. For this reason, we conducted toxicity studies *in vitro* and *in vivo* to investigate the possible undesirable effects of RPLO. RPLO did not alter cell viability or the survival of nematodes exposed to different oil concentrations. The *C. elegans* experimental model has been used to evaluate the possible toxic and pharmacological effects of several substances [70]. Although it is a simpler model than the human organism, several homologous genes and molecular pathways related to development, resistance to different stressors, reproduction, and aging are shared [71, 72]. The results of both the *in vitro* and *in vivo* experiments indicated the safety of the topical application or ingestion of RPLO by the Guarani-Kaiowá people of Mato Grosso do Sul.

## Conclusions

In this study, we present the fatty acid composition of RPLO and, for the first time, describe its antioxidant and healing properties. By confirming the pharmacological properties of RPLO, we intend to share this knowledge not only with the scientific community but also with the Guarani-Kaiowá people to reinforce value to the indigenous TK and prevent this secular knowledge from being lost over time.

## Supporting information

**S1 File.**
(PDF)

**S2 File.**
(DOCX)

## Acknowledgments

The first author is a member of the Guarani-Kaiowá indigenous ethnic group. Authors acknowledgment to Julia Cavalheira Veron and Agostinha Vilharva Cáceres grandparents of Kellen Natalice Vilharva, representatives of all indigenous women, who shared their knowledge with their granddaughter. The authors would like to thank the government agencies that support science in Brazil: Fundação de Apoio ao Desenvolvimento do Ensino, Ciência e Tecnologia do Estado de Mato Grosso do Sul (FUNDECT), Coordenação de Aperfeiçoamento de Pessoal de Nível Superior (CAPES), Conselho Nacional de Desenvolvimento Científico e Tecnológico (CNPq), Financiadora de Estudos e Projetos (Finep) and Pró-Reitoria de Ensino de Pós-Graduação e Pesquisa da UFGD (PROPP-UFGD).

## Author Contributions

**Conceptualization:** Kellen Natalice Vilharva, Caio Fernando Ramalho de Oliveira, Edson Lucas dos Santos, Kely de Picoli Souza.

**Data curation:** Kellen Natalice Vilharva, Caio Fernando Ramalho de Oliveira, Edson Lucas dos Santos, Kely de Picoli Souza.

**Formal analysis:** Kellen Natalice Vilharva, Daniel Ferreira Leite, Helder Freitas dos Santos, Katia Ávila Antunes, Paola dos Santos da Rocha, Jaqueline Ferreira Campos, Claudiane Vilharroel Almeida, Maria Lígia Rodrigues Macedo, Denise Brentan Silva, Caio Fernando Ramalho de Oliveira, Edson Lucas dos Santos, Kely de Picoli Souza.

**Funding acquisition:** Edson Lucas dos Santos, Kely de Picoli Souza.

**Investigation:** Kellen Natalice Vilharva, Daniel Ferreira Leite, Helder Freitas dos Santos, Katia Ávila Antunes, Paola dos Santos da Rocha, Jaqueline Ferreira Campos, Claudiane Vilharroel Almeida, Maria Lígia Rodrigues Macedo, Denise Brentan Silva, Caio Fernando Ramalho de Oliveira, Edson Lucas dos Santos, Kely de Picoli Souza.

**Methodology:** Kellen Natalice Vilharva, Daniel Ferreira Leite, Helder Freitas dos Santos, Katia Ávila Antunes, Paola dos Santos da Rocha, Jaqueline Ferreira Campos, Claudiane Vilharroel Almeida, Maria Lígia Rodrigues Macedo, Denise Brentan Silva, Caio Fernando Ramalho de Oliveira, Edson Lucas dos Santos, Kely de Picoli Souza.

**Project administration:** Kely de Picoli Souza.

**Resources:** Kellen Natalice Vilharva, Caio Fernando Ramalho de Oliveira, Kely de Picoli Souza.

**Supervision:** Caio Fernando Ramalho de Oliveira, Edson Lucas dos Santos, Kely de Picoli Souza.

**Validation:** Paola dos Santos da Rocha, Jaqueline Ferreira Campos, Claudiane Vilharroel Almeida, Denise Brentan Silva, Caio Fernando Ramalho de Oliveira, Edson Lucas dos Santos, Kely de Picoli Souza.

**Visualization:** Maria Lígia Rodrigues Macedo, Caio Fernando Ramalho de Oliveira, Kely de Picoli Souza.

**Writing – original draft:** Kellen Natalice Vilharva, Daniel Ferreira Leite, Helder Freitas dos Santos, Katia Ávila Antunes, Paola dos Santos da Rocha, Jaqueline Ferreira Campos, Claudiane Vilharroel Almeida, Maria Lígia Rodrigues Macedo, Denise Brentan Silva, Caio Fernando Ramalho de Oliveira, Edson Lucas dos Santos, Kely de Picoli Souza.

**Writing – review & editing:** Kellen Natalice Vilharva, Caio Fernando Ramalho de Oliveira, Edson Lucas dos Santos, Kely de Picoli Souza.

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
