## [Decision Letter · Decision Letter 0]

31 Dec 2020

PONE-D-20-36667

Rhynchophorus palmarum Linnaeus (Coleoptera: Curculionidae): Guarani-Kaiowá indigenous knowledge and pharmacological activities

PLOS ONE

Dear Dr. de Picoli Souza,

Thank you for submitting your manuscript to PLOS ONE. After careful consideration, we feel that it has merit but does not fully meet PLOS ONE’s publication criteria as it currently stands. Therefore, we invite you to submit a revised version of the manuscript that addresses the points raised during the review process.

The authors are advised to carefully revise the statistical processing of the data, as described by Reviewer #1. The whole text should be meticulously checked for correct language usage and typograpgic errors. Reviewer #1 has required several additional clarifications, mostly related to the methodology used.

We look forward to receiving your revised manuscript.

Kind regards,

Branislav T. Šiler, Ph.D.

Academic Editor

PLOS ONE

Journal Requirements:

"This work was supported by grants from

 Fundação de Apoio ao Desenvolvimento do Ensino, Ciência e Tecnologia do Estado de Mato

Grosso do Sul (FUNDECT), Coordenação de Aperfeiçoamento de Pessoal de Nível Superior

(CAPES), Conselho Nacional de Desenvolvimento Científico e Tecnológico (CNPq),

Financiadora de Estudos e Projetos (Finep) and Pró-Reitoria de Ensino de Pós-Graduação e

Pesquisa da UFGD (PROPP-UFGD)."

4. We note that Figure 2 in your submission contain map images which may be copyrighted. All PLOS content is published under the Creative Commons Attribution License (CC BY 4.0), which means that the manuscript, images, and Supporting Information files will be freely available online, and any third party is permitted to access, download, copy, distribute, and use these materials in any way, even commercially, with proper attribution. For these reasons, we cannot publish previously copyrighted maps or satellite images created using proprietary data, such as Google software (Google Maps, Street View, and Earth). For more information, see our copyright guidelines: http://journals.plos.org/plosone/s/licenses-and-copyright.

4.1.    You may seek permission from the original copyright holder of Figure 2 to publish the content specifically under the CC BY 4.0 license. 

4.2.    If you are unable to obtain permission from the original copyright holder to publish these figures under the CC BY 4.0 license or if the copyright holder’s requirements are incompatible with the CC BY 4.0 license, please either i) remove the figure or ii) supply a replacement figure that complies with the CC BY 4.0 license. Please check copyright information on all replacement figures and update the figure caption with source information. If applicable, please specify in the figure caption text when a figure is similar but not identical to the original image and is therefore for illustrative purposes only.

Reviewers' comments:

Reviewer's Responses to Questions

**Comments to the Author**

1. Is the manuscript technically sound, and do the data support the conclusions?

Reviewer #1: Yes

Reviewer #2: Yes

2. Has the statistical analysis been performed appropriately and rigorously? 

Reviewer #1: Yes

Reviewer #2: Yes

3. Have the authors made all data underlying the findings in their manuscript fully available?

Reviewer #1: Yes

Reviewer #2: Yes

4. Is the manuscript presented in an intelligible fashion and written in standard English?

Reviewer #1: Yes

Reviewer #2: Yes

5. Review Comments to the Author

Reviewer #1: The manuscript (PONE-D-20-36667) entitled "Rhynchophorus palmarum Linnaeus (Coleoptera: Curculionidae): Guarani-Kaiowá indigenous knowledge and pharmacological activities" is about the therapeutic use of oils obtained from R. palmarum larvae. The authors determined the chemical characteristics of the obtained oil, and, for the first time, are described its antioxidant and healing properties. They paid a special attention to the Guarani-Kaiowá ethnic community and their knowledge of zootherapy. Submitted manuscript contains some valuable information, so, this interesting research deserves attention having in mind a possibility of vanishing of traditional knowledge and practices over time. Also, the presented results may be of some help in the future development of various pharmaceutical products.

Unfortunately, I cannot recommend accepting the manuscript in its present form.

Following are some comments and questions raised during the review of the text.

Significant revision of English scientific writing is needed.

ABSTRACT

Abstract is concise and contains factual information.

INTRODUCTION

In the Introduction, the authors stated the objectives of their work that are supported by appropriate facts.

MATERIALS AND METHODS

The experiments are well designed and performed. But, some parts of Material and Methods section need a bit more details. Specifically, these issues are highlighted:

Obtaining R. palmarum larvae oil

In Supplementary material S1, the method of collecting larvae is already described in detail.

Line 93-94 The authors mentioned that the larvae were collected in Takuara village.

When this procedure was conducted? How many larvae were collected?

Line 98 “...larvae were washed with distilled water, dried with paper towels, weighed, and frozen...“

What were the temperatures that larvae were kept on, until the beginning of the experimental procedure?

Determination of the chemical composition of RPLO

Line 124 “...followed by stirring and incubation of the mixture“

The time of incubation?

Antioxidant activity of RPLO

In formula for DPPH percentage, Abs sample is the absorbance of the sample and Abs control is the absorbance of control? Please, provide clarifications.

Line 149-150 “The results of three experiments performed in triplicate were used to determine the antioxidant activity.“

How many samples were in each experiment?

Antimicrobial activity of RPLO

Line 176 “The experiments were carried out in triplicate. “

How many samples were in each of this triplicates?

Cell culture, viability, and migration

Please, provide clarifications for Abs treated cells and Abs control cells.

Statistical analysis

Parametric statistics should be used when the assumptions of the models (ANOVA) can be reasonably met. A test of homogeneity of variances (for example, Levene test for homogeneity of variances) must show that in all cases variances were not significantly heterogeneous, as well as a Kolmogorov-Smirnov test for normal distribution fitting. So, the authors should test data distribution by, for example, Kolmogorov-Smirnov test and point it out in Statistical analysis section. This is very important because normal distribution is prerequisite for using ANOVA test.

RESULTS

Antimicrobial activity of the RPLO

Please, see the Table 3. Did you mean MIC and MBC, instead of CIM and MBC?

Effect of RPLO on the viability of MRC-5 cells

Maybe, it would be better to rephrase this section. E.g., at given concentrations, cell viability was in what range? Compare to the control, which concentration of RPLO the most changes the viability of MRC-5 cells?

In legend of Fig 4, please mention the number of samples. n= 9?

Effect of RPLO on the migration rate of MRC-5 cells

Line 299 "As shown in Fig 5A…" instead of "As shown in Fig 5…"

Line 302-304 Please, rephrase the sentence. Maybe, something like the following: The cell migration rate was statistically increased by 60% in the wells treated with RPLO compared to the wells receiving the control treatment (Fig. 5B).

Figure 5. Legend should be rewritten. It should be as clear and concise as possible. In addition, please mention the number of samples. Was it 9?

Toxicity of RPLO in a C. elegans model

In legend of Fig 6, please mention the number of samples. n= 10?

DISCUSSION

Please improve this section. The discussion should be clearly and concisely presented. Namely, the same things are repeated in several paragraphs – e.g. see lines 369-371 and 388-391; lines 402-408 and 418-425…

REFERENCES

Please, check this section once again.

Reviewer #2: COMMENTS FOR AUTHORS

Congratulations to a very interesting work. This big effort will certainly influence others to follow. It will be an important reference.

The results and conclusions obtained in this study constitute a clear evidence of the need to conserve and promote the medicinal uses of insects (and animals in general) derived from a deep traditional knowledge.

The paper should definitely be published, but it needs a minor revision. Below you will find a series of specific comments and suggestions, and I have just found some typographical errors, typos that I expose.

Title

Page 10 / line 1… Rhynchophorus palmarum (Linnaeus, 1758) (Coleoptera:...

Abstract

Page 11 / line 24… Brazil. This human community use the…

Page 11 / line 25… snout beetle Rhynchophorus palmarum (Linnaeus, 1758) to...

Page 11 / line 32… against Gram-positive and Gram-negative bacteria that…

Introduction

Page 12 / line 24… Traditional knowledge… the authors mention (they write) this concept numerous times. I think it is possible to use the abbreviation TK from this point on.

Page 12 / line 24… species for their benefit (1). At this point it is possible to include more general and important references; for example:

• Gibson, J. 2016. Community Resources: Intellectual Property, International Trade and Protection of Traditional Knowledge. Routledge, London.

• Robinson, D.F., A. Abdel-Latif, and P. Roffe (eds.) 2017. Protecting Traditional Knowledge: The WIPO Intergovernmental Committee on Intellectual Property and Genetic Resources, Traditional Knowledge and Folklore. Routledge, London.

• Nelson, M.K., and D. Shilling (eds.) 2018. Traditional Ecological Knowledge: Learning from Indigenous Practices for Environmental Sustainability. Cambridge University Press, Cambridge.

Page 12 / line 57… the beetle Rhynchophorus palmarum (Linnaeus, 1758) (Coleoptera:...

Page 12 / line 60… from the integument of R. palmarum larvae is… (not carcass).

Page 12 / line 61… and skin infections (3). At this point it is possible to include more references; for example:

• Cerda, H., R. Martínez, N. Briceno, L. Pizzoferrato, P. Manzi, M. Tommaseo Ponzetta, O. Marín, and M.G. Paoletti, 2001. Palm worm: (Rhynchophorus palmarum) traditional food in Amazonas, Venezuela-nutritional composition, small scale production and tourist palatability. Ecology of Food and Nutrition, 40: 13-32.

• Cartay, R., V. Dimitrov, and M. Feldman, 2020. An insect bad for agriculture but good for human consumption: The case of Rhynchophorus palmarum: A social science perspective. In: Edible Insects, (Mikkola, H., ed.). IntechOpen, DOI: 10.5772/intechopen.87165. Available from: https://www.intechopen.com/books/edible-insects/an-insect-bad-for-agriculture-but-good-for-human-consumption-the-case-of-em-rhynchophorus-palmarum-e

I even think that it is appropriate to bring reference number 16 to this point (Delgado et al. 2019).

Materials and Methods

Page 14 / line 115… at the bottom of Figure 3 appears, in parentheses, the word "arrow" in relation to image C. In this photograph I do not see the arrow.

Page 15… references number 8 (line 120) and 9 (line 141) appear before number 7, which is on page 16 (line 154). Correct this.

Page 15 / line 121… São Paulo

Page 15 / line 140… described by Gupta and Gupta in 2011,…

Page 16 / line 157… São Paulo

Page 17 / line 178… São Paulo

Results

Page 22 / line 272… against Gram-positive and Gram-negative bacteria...

Page 23 / Table 3… the abbreviations MIC and MBC are included in the title of the table… Why do the abbreviations “CIM” and “CBM” appear in the body of the table? Correct this.

Discussion

Page 26 / line 346… isolated from insects is…

Page 26 / lines 347 and 348… change the order of the references: (3) (16), instead of (16) (3).

Page 26 / line 355… with obesity (20), diabetes (21),… enter spaces.

Page 27 / line 362… functions (24-25),… enter a space.

Page 27 / line 366… study by Souza et al. (30), the intake… placed here, this reference is better, at this point (not at the end of the sentence).

Page 28 / line 401… in rats (47-48). … enter a space.

Page 28 / line 408… to sunlight (51-52). … enter a space.

Page 29 / line 414… extracted from the colubrid snake Spilotes pullatus (Linnaeus, 1758) (54),…

Page 29 / line 415… extracted from Boa constrictor Linnaeus, 1758, which…

Page 29 / line 419… for ingestion…

Page 29 / lines 422 and 423… Studies examining the antioxidant activity of oils obtained from insects are scarce in the literature (XXXX);… At this point in the “Discussion” section, I believe that the inclusion of some references is very convenient. I provide you with some examples:

• Ekpo, K.E., A.O. Onigbinde, and I.O. Asia, 2009. Pharmaceutical potentials of the oils of some popular insects consumed in southern Nigeria. African Journal of Pharmacy and Pharmacology, 3(2): 51-57.

• Oonincx, D.G.A.B., S. van Broekhoven, A. van Huis, and J.J.A. van Loon, 2015. Feed conversion, survival and development, and composition of four insect species on diets composed of food by-products. PLoS One, 10(12): e0144601.

• Sosa, D.A.T., and V. Fogliano, 2017. Potential of Insect-Derived Ingredients for Food Applications. In: Insect Physiology and Ecology (Shields, V.D.C., ed.), pp. 215-232. IntechOpen. http://dx.doi.org/10.5772/67318

• Marusich, E., H. Mohamed, Y. Afanasev, and S. Leonov, 2020. Fatty acids from Hermetia illucens larvae fat inhibit the proliferation and growth of actual phytopathogens. Microorganisms 8(9): 1423.

Page 29 / line 431… conducted by Guidoni et al. (46), who… This is not reference number 42. Correct this.

6. PLOS authors have the option to publish the peer review history of their article (what does this mean?). If published, this will include your full peer review and any attached files.

Reviewer #1: No

Reviewer #2: **Yes: **José Antonio González

---

## [Author Response · Author response to Decision Letter 0]

18 Feb 2021

Dear editor,

We are very grateful for the comments raised during the review process. We would like to make it clear that all suggestions have been analyzed and the manuscript has been modified. The responses to each survey were entered as "> response". In the responses, the indicated lines correspond to the file "Revised Article with Changes Highlighted".

The manuscript (PONE-D-20-36667) entitled "Rhynchophorus palmarum Linnaeus (Coleoptera: Curculionidae): Guarani-Kaiowá indigenous knowledge and pharmacological activities" is about the therapeutic use of oils obtained from R. palmarum larvae. The authors determined the chemical characteristics of the obtained oil, and, for the first time, are described its antioxidant and healing properties. They paid a special attention to the Guarani-Kaiowá ethnic community and their knowledge of zootherapy. Submitted manuscript contains some valuable information, so, this interesting research deserves attention having in mind a possibility of vanishing of traditional knowledge and practices over time. Also, the presented results may be of some help in the future development of various pharmaceutical products.

Unfortunately, I cannot recommend accepting the manuscript in its present form.

Following are some comments and questions raised during the review of the text.

Significant revision of English scientific writing is needed.

ABSTRACT

Abstract is concise and contains factual information.

INTRODUCTION

In the Introduction, the authors stated the objectives of their work that are supported by appropriate facts.

MATERIALS AND METHODS 

The experiments are well designed and performed. But, some parts of Material and Methods section need a bit more details. Specifically, these issues are highlighted:

 Obtaining R. palmarum larvae oil

In Supplementary material S1, the method of collecting larvae is already described in detail. 

Line 93-94 The authors mentioned that the larvae were collected in Takuara village. 

 When this procedure was conducted? How many larvae were collected?

Line 98 “...larvae were washed with distilled water, dried with paper towels, weighed, and frozen...“

What were the temperatures that larvae were kept on, until the beginning of the experimental procedure?

> In this version of the manuscript we added further information regard the process of oil obtainment, informing the number of larvae, temperature of storage and further details requested (line 98-99). 

Determination of the chemical composition of RPLO

Line 124 “...followed by stirring and incubation of the mixture“

 The time of incubation?

> We inserted in this version of the manuscript the time of incubation, of 15 min (line 125).

Antioxidant activity of RPLO

In formula for DPPH percentage, Abs sample is the absorbance of the sample and Abs control is the absorbance of control? Please, provide clarifications.

> We appreciate this observation and now, we included more information about the formula (line 148).

Line 149-150 “The results of three experiments performed in triplicate were used to determine the antioxidant activity.“

How many samples were in each experiment?

> This was another important observation raised by referee. We included more details about this topic, informing that three independent assays. In each assay, we used triplicates for each oil concentration (n=3). Thus, each assay was carried out with n= 3. The sum of three independent assays resulted in a total samples of n= 9 (line 152-153).

Antimicrobial activity of RPLO

Line 176 “The experiments were carried out in triplicate. “

How many samples were in each of this triplicates?

> In each assay, we used triplicates for each oil concentration (n=3). Thus, each assay was carried out with n= 3. The sum of three independent assays resulted in a total samples of n= 9 (3 assays using a 3 samples) (line 180).

Cell culture, viability, and migration

Please, provide clarifications for Abs treated cells and Abs control cells.

> We appreciate this observation and now, we included more information about the formula (line 206).

Statistical analysis

Parametric statistics should be used when the assumptions of the models (ANOVA) can be reasonably met. A test of homogeneity of variances (for example, Levene test for homogeneity of variances) must show that in all cases variances were not significantly heterogeneous, as well as a Kolmogorov-Smirnov test for normal distribution fitting. So, the authors should test data distribution by, for example, Kolmogorov-Smirnov test and point it out in Statistical analysis section. This is very important because normal distribution is prerequisite for using ANOVA test.

> We thank the referee regards this so important observation. In this version of the manuscript, we carried out further statistics analyses, such as suggested. So, we included the test of homogeneity of variances (Levene) and for normal distribution (KS test). Since non-significant values were obtained, we continued using ANOVA (line 242-244).

RESULTS 

Antimicrobial activity of the RPLO

Please, see the Table 3. Did you mean MIC and MBC, instead of CIM and MBC?

> These mistakes were corrected.

Effect of RPLO on the viability of MRC-5 cells

Maybe, it would be better to rephrase this section. E.g., at given concentrations, cell viability was in what range? Compare to the control, which concentration of RPLO the most changes the viability of MRC-5 cells?

> This section was rewritten to improve the clarity (line 294-298).

In legend of Fig 4, please mention the number of samples. n= 9?

> The mention regards the n=9 was included (line 302).

Effect of RPLO on the migration rate of MRC-5 cells

Line 299 "As shown in Fig 5A…" instead of "As shown in Fig 5…"

Line 302-304 Please, rephrase the sentence. Maybe, something like the following: The cell migration rate was statistically increased by 60% in the wells treated with RPLO compared to the wells receiving the control treatment (Fig. 5B).

> Based in observation of referee, we modified the section, improving the clarity of suggested sentences (line 306-310).

Figure 5. Legend should be rewritten. It should be as clear and concise as possible. In addition, please mention the number of samples. Was it 9?

> Actually, the legend contained excessive sentences. This mistake was corrected and the number of samples was included in this version (line 314-322).

Toxicity of RPLO in a C. elegans model

In legend of Fig 6, please mention the number of samples. n= 10?

> This was a good observation. This assay contains a n= 90, formed by triplicates containing 10 worms/well (n= 30). Once the assay was carried out three times independently, the number of 90 was obtained (line 332). 

DISCUSSION

Please improve this section. The discussion should be clearly and concisely presented. Namely, the same things are repeated in several paragraphs – e.g. see lines 369-371 and 388-391; lines 402-408 and 418-425…

> We included modifications in discussion topic in order to make more conscious. However, in some points different approaches were necessary. So, whenever possible we improved the section.

REFERENCES

Please, check this section once again.

> We made corrections in references section.

---

## [Editor Report · Decision Letter 1]

19 Feb 2021

PONE-D-20-36667R1

Rhynchophorus palmarum Linnaeus (Coleoptera: Curculionidae): Guarani-Kaiowá indigenous knowledge and pharmacological activities

PLOS ONE

Dear Dr. de Picoli Souza,

Thank you for submitting your manuscript to PLOS ONE. After careful consideration, we feel that it has merit but does not fully meet PLOS ONE’s publication criteria as it currently stands. Therefore, we invite you to submit a revised version of the manuscript that addresses the points raised during the review process.

The authors have revised the manuscript according to the Reviewer #1' report only. The revision based on the reports of both Reviewers are needed. In addition, Response to reviewers should be better structured and authors' reposes have to be provided to each point raised by the Reviewers, instead of being provided for some of them. Cover letter needs to be updated.

We look forward to receiving your revised manuscript.

Kind regards,

Branislav T. Šiler, Ph.D.

Academic Editor

PLOS ONE

---

## [Author Response · Author response to Decision Letter 1]

9 Mar 2021

Dear editor,

We are very grateful for the comments raised during the review process. We would like to make it clear that all suggestions have been analyzed and the manuscript has been modified. The responses to each survey were entered as ‘> response’.

Reviewer #1: COMMENTS FOR AUTHORS

The manuscript (PONE-D-20-36667) entitled "Rhynchophorus palmarum Linnaeus (Coleoptera: Curculionidae): Guarani-Kaiowá indigenous knowledge and pharmacological activities" is about the therapeutic use of oils obtained from R. palmarum larvae. The authors determined the chemical characteristics of the obtained oil, and, for the first time, are described its antioxidant and healing properties. They paid a special attention to the Guarani-Kaiowá ethnic community and their knowledge of zootherapy. Submitted manuscript contains some valuable information, so, this interesting research deserves attention having in mind a possibility of vanishing of traditional knowledge and practices over time. Also, the presented results may be of some help in the future development of various pharmaceutical products.

Unfortunately, I cannot recommend accepting the manuscript in its present form.

Following are some comments and questions raised during the review of the text.

Significant revision of English scientific writing is needed.

ABSTRACT

Abstract is concise and contains factual information.

INTRODUCTION

In the Introduction, the authors stated the objectives of their work that are supported by appropriate facts.

MATERIALS AND METHODS 

The experiments are well designed and performed. But, some parts of Material and Methods section need a bit more details. Specifically, these issues are highlighted:

 Obtaining R. palmarum larvae oil

In Supplementary material S1, the method of collecting larvae is already described in detail. 

Line 93-94 The authors mentioned that the larvae were collected in Takuara village. 

 When this procedure was conducted? How many larvae were collected?

Line 98 “...larvae were washed with distilled water, dried with paper towels, weighed, and frozen...“

What were the temperatures that larvae were kept on, until the beginning of the experimental procedure?

> In this version of the manuscript we added further information regard the process of oil obtainment, informing the number of larvae, temperature of storage and further details requested. 

Determination of the chemical composition of RPLO

Line 124 “...followed by stirring and incubation of the mixture“

 The time of incubation?

> We inserted in this version of the manuscript the time of incubation (15 min).

Antioxidant activity of RPLO

In formula for DPPH percentage, Abs sample is the absorbance of the sample and Abs control is the absorbance of control? Please, provide clarifications.

> We appreciate this observation and now, we included more information about the formula.

Line 149-150 “The results of three experiments performed in triplicate were used to determine the antioxidant activity.“

How many samples were in each experiment?

> This was another important observation raised by referee. We included more details about this topic, informing that three independent assays. In each assay, we used triplicates for each oil concentration (n=3). Thus, each assay was carried out with n= 3. The sum of three independent assays resulted in a total samples of n= 9.

Antimicrobial activity of RPLO

Line 176 “The experiments were carried out in triplicate. “

How many samples were in each of this triplicates?

> In each assay, we used triplicates for each oil concentration (n=3). Thus, each assay was carried out with n= 3. The sum of three independent assays resulted in a total samples of n= 9 (3 assays using a 3 samples)

Cell culture, viability, and migration

Please, provide clarifications for Abs treated cells and Abs control cells.

> We appreciate this observation and now, we included more information about the formula.

Statistical analysis

Parametric statistics should be used when the assumptions of the models (ANOVA) can be reasonably met. A test of homogeneity of variances (for example, Levene test for homogeneity of variances) must show that in all cases variances were not significantly heterogeneous, as well as a Kolmogorov-Smirnov test for normal distribution fitting. So, the authors should test data distribution by, for example, Kolmogorov-Smirnov test and point it out in Statistical analysis section. This is very important because normal distribution is prerequisite for using ANOVA test.

> We thank the referee regards this so important observation. In this version of the manuscript, we carried out further statistics analyses, such as suggested. So, we included the test of homogeneity of variances (Levene) and for normal distribution (KS test). Since non-significant values were obtained, we continued using ANOVA.

RESULTS 

Antimicrobial activity of the RPLO

Please, see the Table 3. Did you mean MIC and MBC, instead of CIM and MBC?

> These mistakes were corrected.

Effect of RPLO on the viability of MRC-5 cells

Maybe, it would be better to rephrase this section. E.g., at given concentrations, cell viability was in what range? Compare to the control, which concentration of RPLO the most changes the viability of MRC-5 cells?

> This section was rewritten to improve the clarity.

In legend of Fig 4, please mention the number of samples. n= 9?

> The mention regards the n=9 was included.

Effect of RPLO on the migration rate of MRC-5 cells

Line 299 "As shown in Fig 5A…" instead of "As shown in Fig 5…"

Line 302-304 Please, rephrase the sentence. Maybe, something like the following: The cell migration rate was statistically increased by 60% in the wells treated with RPLO compared to the wells receiving the control treatment (Fig. 5B).

> Based in observation of referee, we modified the section, improving the clarity of suggested sentences.

Figure 5. Legend should be rewritten. It should be as clear and concise as possible. In addition,

please mention the number of samples. Was it 9?

> Actually, the legend contained excessive sentences. This mistake was corrected, and the number of samples was included in this version.

Toxicity of RPLO in a C. elegans model

In legend of Fig 6, please mention the number of samples. n= 10?

> This was a good comment. This assay contains a n= 90, formed by triplicates containing 10 worms/well (n= 30). Once the assay was carried out three times independently, the number of 90 was obtained. 

DISCUSSION

Please improve this section. The discussion should be clearly and concisely presented. Namely, the same things are repeated in several paragraphs – e.g. see lines 369-371 and 388-391; lines 402-408 and 418-425…

> We included modifications in discussion topic in order to make more conscious. However, in some points different approaches were necessary. So, whenever possible we improved the section.

REFERENCES

Please, check this section once again.

> We made corrections and addition of further references in this version.

Reviewer #2: COMMENTS FOR AUTHORS

Congratulations to a very interesting work. This big effort will certainly influence others to follow. It will be an important reference. The results and conclusions obtained in this study constitute a clear evidence of the need to conserve and promote the medicinal uses of insects (and animals in general) derived from a deep traditional knowledge. The paper should definitely be published, but it needs a minor revision. Below you will find a series of specific comments and suggestions, and I have just found some typographical errors, typos that I expose.

Title

Page 10 / line 1… Rhynchophorus palmarum (Linnaeus, 1758) (Coleoptera:...

> We included in this version the correct taxonomic mention to C. Linnaeus, 1758.

Abstract

Page 11 / line 24… Brazil. This human community use the…

Page 11 / line 25… snout beetle Rhynchophorus palmarum (Linnaeus, 1758) to...

Page 11 / line 32… against Gram-positive and Gram-negative bacteria that…

> Regards the three mentions, we would like to keep the first one (line 24), since the first author of the manuscript is an indigenous researcher woman. For this reason, the expression ‘my people’ was considered pertinent. The changes in lines 25 and 32 were carried out.

Introduction

Page 12 / line 24… Traditional knowledge… the authors mention (they write) this concept numerous times. I think it is possible to use the abbreviation TK from this point on.

> We appreciate the important observation. Along the manuscript we used the abbreviation TK replacing the expression Traditional knowledge.

Page 12 / line 24… species for their benefit (1). At this point it is possible to include more general and important references; for example:

• Gibson, J. 2016. Community Resources: Intellectual Property, International Trade and Protection of Traditional Knowledge. Routledge, London.

• Robinson, D.F., A. Abdel-Latif, and P. Roffe (eds.) 2017. Protecting Traditional Knowledge: The WIPO Intergovernmental Committee on Intellectual Property and Genetic Resources, Traditional Knowledge and Folklore. Routledge, London.

• Nelson, M.K., and D. Shilling (eds.) 2018. Traditional Ecological Knowledge: Learning from Indigenous Practices for Environmental Sustainability. Cambridge University Press, Cambridge.

> In this version of the manuscript these important references were included to improve the sources of information regards Traditional Knowledge (TK).

Page 12 / line 57… the beetle Rhynchophorus palmarum (Linnaeus, 1758) (Coleoptera:...

Page 12 / line 60… from the integument of R. palmarum larvae is… (not carcass).

Page 12 / line 61… and skin infections (3). At this point it is possible to include more references; for example:

• Cerda, H., R. Martínez, N. Briceno, L. Pizzoferrato, P. Manzi, M. Tommaseo Ponzetta, O. Marín, and M.G. Paoletti, 2001. Palm worm: (Rhynchophorus palmarum) traditional food in Amazonas, Venezuela-nutritional composition, small scale production and tourist palatability. Ecology of Food and Nutrition, 40: 13-32.

• Cartay, R., V. Dimitrov, and M. Feldman, 2020. An insect bad for agriculture but good for human consumption: The case of Rhynchophorus palmarum: A social science perspective. In: Edible Insects, (Mikkola, H., ed.). IntechOpen, DOI: 10.5772/intechopen.87165. Available from: https://www.intechopen.com/books/edible-insects/an-insect-bad-for-agriculture-but-good-for-human-consumption-the-case-of-em-rhynchophorus-palmarum-e

I even think that it is appropriate to bring reference number 16 to this point (Delgado et al. 2019).

> We also appreciate the suggestions in order to improve the final quality of introduction section, including additional references suggested.

Materials and Methods

Page 14 / line 115… at the bottom of Figure 3 appears, in parentheses, the word "arrow" in relation to image C. In this photograph I do not see the arrow.

Page 15… references number 8 (line 120) and 9 (line 141) appear before number 7, which is on page 16 (line 154). Correct this.

Page 15 / line 121… São Paulo

Page 15 / line 140… described by Gupta and Gupta in 2011,…

Page 16 / line 157… São Paulo

Page 17 / line 178… São Paulo

> In this version, the Figure 3 mentioned before is Figure 2, since the map image was removed. The legend was corrected, as well the further points raised in lines 121, 140, 157, and 178.

Results

Page 22 / line 272… against Gram-positive and Gram-negative bacteria...

Page 23 / Table 3… the abbreviations MIC and MBC are included in the title of the table… Why do the abbreviations “CIM” and “CBM” appear in the body of the table? Correct this.

> The changes suggested by reviewer were accepted and the corrections in abbreviations CIM and CBM by MIC and MBC.

Discussion

Page 26 / line 346… isolated from insects is…

Page 26 / lines 347 and 348… change the order of the references: (3) (16), instead of (16) (3).

Page 26 / line 355… with obesity (20), diabetes (21),… enter spaces.

Page 27 / line 362… functions (24-25),… enter a space.

Page 27 / line 366… study by Souza et al. (30), the intake… placed here, this reference is better, at this point (not at the end of the sentence).

Page 28 / line 401… in rats (47-48). … enter a space.

Page 28 / line 408… to sunlight (51-52). … enter a space.

Page 29 / line 414… extracted from the colubrid snake Spilotes pullatus (Linnaeus, 1758) (54),…

Page 29 / line 415… extracted from Boa constrictor Linnaeus, 1758, which…

Page 29 / line 419… for ingestion…

> All these observations are pertinent, for this reason the mentioned lines were changed according referee’s suggestions.

Page 29 / lines 422 and 423… Studies examining the antioxidant activity of oils obtained from insects are scarce in the literature (XXXX);… At this point in the “Discussion” section, I believe that the inclusion of some references is very convenient. I provide you with some examples:

• Ekpo, K.E., A.O. Onigbinde, and I.O. Asia, 2009. Pharmaceutical potentials of the oils of some popular insects consumed in southern Nigeria. African Journal of Pharmacy and Pharmacology, 3(2): 51-57.

• Oonincx, D.G.A.B., S. van Broekhoven, A. van Huis, and J.J.A. van Loon, 2015. Feed conversion, survival and development, and composition of four insect species on diets composed of food by-products. PLoS One, 10(12): e0144601.

• Sosa, D.A.T., and V. Fogliano, 2017. Potential of Insect-Derived Ingredients for Food Applications. In: Insect Physiology and Ecology (Shields, V.D.C., ed.), pp. 215-232. IntechOpen. http://dx.doi.org/10.5772/67318

• Marusich, E., H. Mohamed, Y. Afanasev, and S. Leonov, 2020. Fatty acids from Hermetia illucens larvae fat inhibit the proliferation and growth of actual phytopathogens. Microorganisms 8(9): 1423.

Page 29 / line 431… conducted by Guidoni et al. (46), who… This is not reference number 42. Correct this.

> In this version we fixed this mistake.

---

## [Decision Letter · Decision Letter 2]

18 Mar 2021

PONE-D-20-36667R2

Rhynchophorus palmarum (Linnaeus, 1758) (Coleoptera: Curculionidae): Guarani-Kaiowá indigenous knowledge and pharmacological activities

PLOS ONE

Dear Dr. de Picoli Souza,

Thank you for submitting your manuscript to PLOS ONE. After careful consideration, we feel that it has merit but does not fully meet PLOS ONE’s publication criteria as it currently stands. Therefore, we invite you to submit a revised version of the manuscript that addresses the points raised during the review process.

Some additional work on statistical processing is needed as stated in the Reviewer's #2 report (provided in the attachment). Figure captions also need the authors' attention. Please put references in the text into square brackets.

We look forward to receiving your revised manuscript.

Kind regards,

Branislav T. Šiler, Ph.D.

Academic Editor

PLOS ONE

Journal Requirements:

Reviewers' comments:

Reviewer's Responses to Questions

**Comments to the Author**

1. If the authors have adequately addressed your comments raised in a previous round of review and you feel that this manuscript is now acceptable for publication, you may indicate that here to bypass the “Comments to the Author” section, enter your conflict of interest statement in the “Confidential to Editor” section, and submit your "Accept" recommendation.

Reviewer #1: (No Response)

Reviewer #2: All comments have been addressed

2. Is the manuscript technically sound, and do the data support the conclusions?

Reviewer #1: (No Response)

Reviewer #2: Yes

3. Has the statistical analysis been performed appropriately and rigorously? 

Reviewer #1: (No Response)

Reviewer #2: Yes

4. Have the authors made all data underlying the findings in their manuscript fully available?

Reviewer #1: (No Response)

Reviewer #2: Yes

5. Is the manuscript presented in an intelligible fashion and written in standard English?

Reviewer #1: (No Response)

Reviewer #2: Yes

6. Review Comments to the Author

Reviewer #1: (No Response)

Reviewer #2: I want to express my congratulations to the authors for the effort made to review and complete the article. The final version is truly satisfying. My recommendation is that it can be definitely published in the journal PLOS ONE.

Best regards and stay healthy.

7. PLOS authors have the option to publish the peer review history of their article (what does this mean?). If published, this will include your full peer review and any attached files.

Reviewer #1: No

Reviewer #2: **Yes: **José Antonio González

---

## [Author Response · Author response to Decision Letter 2]

25 Mar 2021

The manuscript PONE-D-20-36667R2 entitled “Rhynchophorus palmarum (Linnaeus, 1758) (Coleoptera: Curculionidae): Guarani Kaiowá indigenous knowledge and pharmacological activities” was reviewed and the authors have addressed all the issues that reviewers had raised. I suggest that the manuscript should be accepted after an additional minor revision.

We would like to thank Dr. José Antonio González for taking care of the smallest details that will certainly enrich our manuscript. Below are the responses to the surveys, which are marked in blue.

Minor comment:

The Material and Methods section has been greatly improved. But, Statistical analysis description could be a bit more precise, something like this: 

The data were expressed as mean ± standard error of the mean (SEM). Prior to the statistical analysis, Levene´s test of homogeneity of variances was used to assume that variances are equal across groups or samples and Kolmogorov-Smirnov test was used to assess the normality of the data. The mean values of the cell migration rate were analyzed by One-way analysis of variance (ANOVA) followed by Dunnett's post-test. In in vivo assays, a t-test was used to determine differences between groups. All analyses were performed using GraphPad Prism 5 software. The results were considered to be statistically significant at p ≤ 0.05.

> We appreciate the suggestion, in this version the manuscript the section Statistical analysis was rewritten observing the improvement of clarity.

Also, Figure legends would be rewritten, e.g.

Fig 3. Viability of MRC-5 cells incubated with different concentrations of RPLO for 24 h. The graph shows the means ± SEM of three independent experiments performed in triplicate (n= 9). The observed changes were not statistically significant (t-test).

Fig 4. The effect of RPLO on the migration rate of MRC-5 cells. (A) Representative image from assay. The wound areas were determined at 0 h and after 24 h of incubation. Control cells were incubated with culture media. Cells incubated with 0.5% of oil were named RPLO. Scale bar is 1 mm. (B) The graph shows the means ± SEM of three independent experiments performed in triplicate (n= 9). An asterisk (*) indicate statistically significant differences between mean values (one-way ANOVA, Dunnett's post-test, p <0.05). 

Fig 5. Acute toxicity of RPLO in the C. elegans experimental model in vivo. Viability was recorded at (A) 24 h and (B) 48 h of treatment. The graph shows the means ± SEM of three independent experiments performed in triplicate (n= 90). The observed changes were not statistically significant (t-test).

> The same care regards the previous appointment was took in consideration, in order to become the Figure legends clear.

---

## [Editor Report · Decision Letter 3]

29 Mar 2021

Rhynchophorus palmarum (Linnaeus, 1758) (Coleoptera: Curculionidae): Guarani-Kaiowá indigenous knowledge and pharmacological activities

PONE-D-20-36667R3

Dear Dr. de Picoli Souza,

We’re pleased to inform you that your manuscript has been judged scientifically suitable for publication and will be formally accepted for publication once it meets all outstanding technical requirements.

Kind regards,

Branislav T. Šiler, Ph.D.

Academic Editor

PLOS ONE
---

## [Editor Report · Acceptance letter]

6 Apr 2021

PONE-D-20-36667R3 

*Rhynchophorus palmarum* (Linnaeus, 1758) (Coleoptera: Curculionidae): Guarani-Kaiowá indigenous knowledge and pharmacological activities 

Dear Dr. de Picoli Souza:

I'm pleased to inform you that your manuscript has been deemed suitable for publication in PLOS ONE. Congratulations! Your manuscript is now with our production department. 

Kind regards, 

on behalf of

Dr. Branislav T. Šiler 

Academic Editor

PLOS ONE